# Morphological and DNA sequence data uncover a new millipede species in the *Thyropygus opinatus* subgroup and assign *T. peninsularis* to this subgroup (Diplopoda: Spirostreptida: Harpagophoridae)

Piyatida Pimvichai[1,2], Henrik Enghoff[3], Karin Breugelmans[2], Brigitte Segers[2] and Thierry Backeljau[2,4]

[1] Department of Biology, Faculty of Science, Mahasarakham University, Maha Sarakham, Thailand
[2] OD Taxonomy and Phylogeny, Royal Belgian Institute of Natural Sciences, Brussels, Belgium
[3] Natural History Museum of Denmark, University of Copenhagen, Copenhagen, Denmark
[4] Evolutionary Ecology Group, University of Antwerp, Antwerp, Belgium

Corresponding author
Piyatida Pimvichai,
piyatida.p@msu.ac.th

## ABSTRACT

The millipede genus *Thyropygus* Pocock, 1894 is one of the most diverse genera within the family Harpagophoridae in Southeast Asia. The *Thyropygus opinatus* subgroup, belonging to the *T. allevatus* group, is distinguished by the presence of an additional projection on the anterior coxal fold. Here, we describe a new species of the *T. opinatus* subgroup, *Thyropygus payamense* sp. nov., from Payam Island, Ranong Province, Thailand, based on morphological and DNA sequence data. The mean interspecific COI divergence between the new species and other *Thyropygus* species is 0.13 ± 0.02 (range: 0.07–0.16). The new species is distinguished by (1) a small, slender, pointed spine at base of femoral spine, (2) a short, triangular mesal process of the anterior coxal fold, and (3) a short, slender, slightly mesad-curving tibial spine. Additionally, *T. peninsularis* Hoffman, 1982 is confirmed as a member of the *T. opinatus* subgroup, because it shares key gonopodal characters with other species in this subgroup, while COI and 16S rRNA sequence data firmly support this new classification, with a mean interspecific COI sequence divergence of 0.13 ± 0.03 (range: 0.07–0.17) from other species in the *T. allevatus* group. An identification key for all 29 species in the *T. opinatus* subgroup is provided. Further research is needed to assess the taxonomic status of, and phylogenetic relationships within, this subgroup, which, except for two species, may tentatively represent an endemic species radiation in the peninsular area of Thailand, Malaysia and Myanmar.

## INTRODUCTION

The millipede genus *Thyropygus* Pocock, 1894 is widely distributed across Thailand and Southeast Asia and currently comprises 67 recognized species, 46 of which are
exclusively found in Thailand (*Pimvichai, Enghoff & Backeljau, 2023*). Most Thai species belong to the informal *T. allevatus* group, which was defined by *Hoffman (1975)* on the basis of two features of the gonopod telopodite: (1) the presence of tibial and femoral spines, and (2) the tibial spine being very long and recurved proximad towards the femoral spine. The *T. allevatus* group is widely distributed throughout Thailand, Vietnam, Laos, Cambodia, and Peninsular Malaysia (*Enghoff, 2005*). By combining morphological and DNA sequence data, the *T. allevatus* group has been further divided into four informal subgroups: (1) the *T. opinatus* subgroup, (2) the *T. induratus* subgroup, (3) the *T. cuisinieri* subgroup, and (4) the *T. allevatus* subgroup (*Pimvichai, Enghoff & Panha, 2009a*; *Pimvichai, Enghoff & Panha, 2009b*; *Pimvichai, Enghoff & Panha, 2011a*; *Pimvichai, Enghoff & Panha, 2011b*; *Pimvichai, Enghoff & Panha, 2014*; *Pimvichai et al., 2016*; *Pimvichai, Enghoff & Backeljau, 2023*). Within this system, the *T. opinatus* subgroup is characterized by the presence of an additional projection on the anterior coxal fold (*Pimvichai et al., 2016*). The *T. opinatus* subgroup is primarily distributed in Thailand, with only two species that also occur outside Thailand: *T. implicatus* (Demange, 1961) in Peninsular Malaysia and *T. opinatus* (Karsch, 1881) in southern Myanmar (*Pimvichai, Enghoff & Panha, 2009a*; *Pimvichai, Enghoff & Panha, 2009b*; *Pimvichai, Enghoff & Panha, 2014*).

Hitherto, the informal subgroup division of the *T. allevatus* group appeared well-supported by the overall congruence between morphological and DNA sequence data. Yet, recently this congruence was challenged with the discovery of two *Thyropygus* species, *T. panhai* Pimvichai, Enghoff & Backeljau, 2023 and *T. somsaki* Pimvichai, Enghoff & Backeljau, 2023, that morphologically clearly belong to the *T. induratus* subgroup, but whose COI sequences do not support this assignment. In fact, including both species in the COI phylogeny made that the monophyly of the *T. induratus* subgroup was no longer supported (*Pimvichai, Enghoff & Backeljau, 2023*). Hence extended taxon sampling is important to further explore the congruence between morphological and DNA sequence data, and eventual taxonomic validity, of the informal subgroups within the *T. allevatus* group.

Against this background, recently collected millipede specimens from Payam Island in the Andaman Sea appeared morphologically to belong to a new species of the *T. opinatus* subgroup, thus offering an opportunity to test the consistency of this subgroup. The present contribution aims to do so by formally describing and DNA barcoding this new species as *Thyropygus payamense* sp. nov. In addition, it provides an updated morphological identification key of all species currently assigned to the *T. opinatus* subgroup and discusses the taxonomic position of *T. peninsularis Hoffman, 1982*, a species which until recently was assigned to the *T. erythropleurus* group (*Hoffman, 1982*; *Pimvichai, Enghoff & Panha, 2009a*), but whose transfer to the *T. opinatus* subgroup in the *T. allevatus* group (*Pimvichai, Enghoff & Backeljau, 2023*) is here formally confirmed.

## MATERIALS & METHODS

### Specimen collection

In November 2022 live specimens of the new species were hand-collected at Payam Island, Ranong Province, Thailand and preserved in 70% ethanol ($n = 3$) or stored in a freezer at $-20\ °C$ ($n = 10$). This material has been deposited in the collections of the Museum of Zoology, Chulalongkorn University, Bangkok, Thailand (CUMZ). Another specimen of *T. payamense* sp. nov. from Payam Island, collected in April 2013 by J. Urbanski and preserved in 70% ethanol, is kept in the Natural History Museum of Denmark (NHMD).

This research was conducted under the approval of the Animal Care and Use regulations (numbers U1-07304-2560 and IACUC-MSU-037/2019) of the National Research Council of Thailand.

The electronic version of this article in Portable Document Format (PDF) will represent a published work according to the International Commission on Zoological Nomenclature (ICZN), and hence the new names contained in the electronic version are effectively published under that Code from the electronic edition alone. This published work and the nomenclatural acts it contains have been registered in ZooBank, the online registration system for the ICZN. The ZooBank LSIDs (Life Science Identifiers) can be resolved and the associated information viewed through any standard web browser by appending the LSID to the prefix http://zoobank.org/. The LSID for this publication is: urn:lsid:zoobank.org:pub:68E7FD7F-A8E3-4BE9-9B4B-136CDEEBEA88. The online version of this work is archived and available from the following digital repositories: PeerJ, PubMed Central SCIE and CLOCKSS.

### Morphology

Gonopods were photographed with a digital camera and drawings were made using a stereomicroscope and photographs. Gonopod terminology of the *T. opinatus* subgroup follows *Pimvichai, Enghoff & Panha (2009a)*; *Pimvichai, Enghoff & Panha (2009b)*; *Pimvichai et al. (2016)*. A new term is marked in bold:

*ac* = anterior coxal fold: the main part of gonopod in anterior view; confusingly called *posterior* coxal fold by *Demange (1961)* and *Hoffman (1975)*

*aip* = additional spine-like process: between lateral and mesal processes of anterior coxal fold

*alp* = lateral process of anterior coxal fold: the distolateral part of the anterior coxal fold

*amp* = mesal process of anterior coxal fold: an additional projection on the anterior coxal fold, protruding from its mesal margin

*bp* = blepharochaete (pl. -ae): the normal form of apical setae, long, slender, stiffened, and usually pigmented, somewhat reminiscent of the mammalian eyelash (*Hoffman, 1975*)

*cr* = longitudinal crest in gutter of palette: a crest which runs along the middle of the gutter near the tip of the palette

*fe* = femoral spine (also *fe 1* and *fe 2*): a usually long, curved spine on the telopodite, originating slightly distal to the point where the telopodite emerges from the coxa

*lc* = longitudinal crest: a strong longitudinal crest at the mesal margin of *amp* in posterior view

*ll* = lamellar lobe: a small, slightly folded lobe at the basis of the apical part of the telopodite

*lo* = telopodite lobe: a protruding lobe on the telopodite, distal to *fe*

*pa* = palette: the distalmost lobe of the apical part, carrying the row of blepharochaetae

*pc* = posterior coxal fold: the main part of gonopod in posterior view, usually shorter than *ac* and forming a shelf for accommodation of telopodite shaft

*plp* = lateral process of posterior coxal fold: the lateral part of the posterior coxal fold, usually digitiform

*pmp* = mesal process of posterior coxal fold: the mesal part of the posterior coxal fold, usually forming a shelf for accommodation of telopodite shaft

*px* = paracoxite: the basal, lateral part of the posterior coxal fold

**sfe = small spine at the base of femoral spine**: an additional small, slender, sharp spine at the base of femoral spine

*sl* = spatulate lobe: a distinct distal, separate lobe at the apical part, spatulate, sometimes with a distal spine-like process

*sls* = slender long spine: an additional slender long spine (much longer than *ss*) at the base of the apical part of telopodite in posterior view

*ss* = small spine: an additional small spine at the base of the apical part of telopodite in posterior view

*st* = sternum: a small, usually triangular sclerite between the basal parts of the anterior coxal folds

*ti* = tibial spine: a usually long spine on the telopodite, originating distal to the femoral spine, at the basis of the apical part of the telopodite, usually curved in the opposite direction of the femoral spine, the two together forming a circle

Apical part: the part of the telopodite distal to the tibial spine

Shelf: the distal surface of the posterior coxal fold.

### DNA extraction, amplification, and sequencing

Total genomic DNA was extracted from legs of three specimens using the NucleoSpin Tissue kit (Macherey-Nagel, Düren, Germany) following the manufacturer's instructions. PCR amplifications and sequencing of the standard mitochondrial COI DNA barcoding fragment (*Hebert et al., 2003*) and a mitochondrial 16S rRNA fragment were done as described by *Pimvichai et al. (2020)*. The COI fragment was amplified with the primers LCO-1490 and HCO-2198 (*Folmer et al., 1994*), and the 16S rRNA fragment was amplified with the primers 16Sar and 16Sbr (*Kessing et al., 2004*). The new COI and 16S rRNA sequences have been deposited in GenBank under accession numbers PV019345–PV019347 and PV029246–PV029247. Sample data and voucher codes are provided in Table S1.

### DNA sequence analysis

The COI dataset comprised 61 specimens of 33 nominal *Thyropygus* species and four outgroup species from the harpagophorid subfamily Rhynchoproctinae *viz.*, *Anurostreptus barthelemyae* Demange, 1961, *A. sculptus* Demange, 1961, *Armatostreptus armatus* (Demange, 1983), and *Heptischius lactuca* Pimvichai, Enghoff & Panha, 2010 (Table S1). The same specimens were used for the 16S rRNA and combined COI + 16S rRNA

datasets, except for *T. payamense* sp. nov. (KPYR3), *T. panhi* and *T. somsaki*, of which no 16S rRNA sequences could be obtained.

Sequence assembly and editing were performed using CodonCode Aligner (ver. 4.0.4; CodonCode Corporation) to combine forward and reverse reads, identify errors, and resolve ambiguities. All sequences were verified using the Basic Local Alignment Search Tool (BLAST, NCBI) and compared against reference sequences in GenBank. Sequence alignment was conducted using MUSCLE (ver. 3.6; *Edgar, 2004*; http://www.drive5.com/muscle). The sequences were evaluated for ambiguous nucleotide sites, saturation, and phylogenetic signal using DAMBE (ver. 5.2.65; *Xia, 2018*; https://dambe.bio.uottawa.ca/DAMBE/dambe.aspx). MEGA11 (ver. 11.0.10; *Tamura, Stecher & Kumar, 2021*; http://www.megasoftware.net) was used to: (1) screen for stop codons, (2) translate nucleotide sequences into amino acids, and (3) calculate uncorrected pairwise p-distances among sequences.

### Phylogenetic analysis

Phylogenetic trees were constructed using maximum likelihood (ML) and Bayesian Inference (BI) approaches.

ML trees were inferred using RAxML (ver. 8.2.12; *Stamatakis, 2014*; http://www.phylo.org/index.php/tools/raxmlhpc2_tgb.html) *via* the CIPRES Science Gateway (*Miller, Pfeiffer & Schwartz, 2010*) and applying the GTR+G substitution model.

BI trees were constructed using MrBayes (ver. 3.2.7a; *Huelsenbeck & Ronquist, 2001*; http://www.phylo.org/index.php/tools/mrbayes_xsede.html). Substitution models were selected using jModeltest (ver. 2.1.10; *Darriba et al., 2012*; https://www.github.com/ddarriba/modeltest2/releases), with the Akaike Information Criterion (*Akaike, 1973*) as the selection criterion. The GTR+I+G model was identified as the best fit model for COI (lnL = 11,936.7043, gamma shape = 0.8820), 16S rRNA (–lnL = 8,382.4103, gamma shape = 0.8950), and the combined COI + 16S rRNA dataset (–lnL = 3,392.4942, gamma shape = 0.4530). BI analyses were run for 10 million (combined dataset), 20 million (COI), and 2 million (16S rRNA) generations. The heating parameter was set to 0.01 for all datasets, and trees were sampled every 1,000 generations. Convergence was confirmed by ensuring that the standard deviation of split frequencies was <0.01. The first 1,000 trees were discarded as burn-in, and the final consensus tree was generated from the last 15,002 (combined dataset), 30,002 (COI), and 3,002 (16S rRNA) trees.

Node support was evaluated using posterior probabilities (PP) for BI and bootstrap values (BV) for ML (based on 1,000 replicates). Nodes with BV ≥ 70% or PP ≥ 0.95 were considered well-supported, while BV <70% or PP <0.95 were considered as poorly supported (*Hillis & Bull, 1993*; *San Mauro & Agorreta, 2010*).

## RESULTS

### DNA sequence data and phylogeny

The uncorrected p-distances between the COI sequences (660 bp) of *Thyropygus* specimens included in this study ranged from 0.00 to 0.18 (Table S2). The mean intraspecific sequence divergence within the *T. allevatus* group was 0.06 ± 0.03 (range: 0.00–0.12).

Mean intraspecific divergence values for individual species of this group were: *T. allevatus* (two specimens) = 0.00; *T. induratus* = 0.05 ± 0.02 (range: 0.02–0.07); *T. payamense* sp. nov. (three specimens) = 0.01 ± 0.02 (range: 0.00–0.01); *T. resimus* = 0.06 ± 0.04 (range: 0.00–0.10); and *T. uncinatus* = 0.06 ± 0.03 (range: 0.00–0.12). The mean interspecific sequence divergence within the *T. allevatus* group (all subgroups included) was 0.14 ± 0.02 (range: 0.02–0.18). The mean interspecific sequence divergence within the *T. opinatus* subgroup was 0.12 ± 0.03 (range: 0.02–0.17). The mean interspecific sequence divergence in the *T. opinatus* subgroup without *T. payamense* sp. nov. = 0.12 ± 0.03 (range: 0.02–0.17). The mean interspecific sequence divergence of *T. payamense* sp. nov. *vs* other species in the *T. opinatus* subgroup = 0.11 ± 0.02 (range: 0.07–0.15). The mean interspecific sequence divergence of *T. payamense* sp. nov. *vs* other species in the *T. allevatus* group = 0.13 ± 0.02 (range: 0.07–0.16).

The uncorrected p-distances between the 16S rRNA sequences (487 bp) of *Thyropygus* species ranged from 0.00 to 0.13 (Table S3). The mean intraspecific sequence divergence within the *T. allevatus* group was 0.02 ± 0.02 (range: 0.00–0.08). Mean intraspecific divergence values for individual species of this group were: *T. allevatus* (two specimens) = 0.00; *T. induratus* = 0.03 ± 0.03 (range: 0.01–0.08); *T. payamense* sp. nov. (two specimens) = 0.00; *T. resimus* = 0.02 ± 0.01 (range: 0.00–0.03); and *T. uncinatus* = 0.02 ± 0.01 (range: 0.00–0.04). The mean interspecific sequence divergence within the *T. allevatus* group (all subgroups included) was 0.08 ± 0.02 (range: 0.00–0.13). The mean interspecific sequence divergence within the *T. opinatus* subgroup was: 0.05 ± 0.02 (range: 0.00–0.9). The mean interspecific sequence divergence in the *T. opinatus* subgroup without *T. payamense* sp. nov. = 0.05 ± 0.02 (range: 0.00–0.09). The mean interspecific sequence divergence of *T. payamense* sp. nov. *vs* other species in the *T. opinatus* subgroup = 0.05 ± 0.02 (range: 0.01–0.08). The mean interspecific sequence divergence of *T. payamense* sp. nov. *vs* other species in the *T. allevatus* group = 0.08 ± 0.03 (range: 0.01–0.12).

The uncorrected p-distances between the sequences of *Thyropygus* species in the combined dataset (COI + 16S rRNA, 1,147 bp) ranged from 0.01 to 0.15 (Table S4). The mean intraspecific sequence divergence within the *T. allevatus* group was 0.04 ± 0.02 (range: 0.00–0.08). Mean intraspecific divergence values for individual species of this group were: *T. allevatus* (two specimens) = 0.00; *T. induratus* = 0.04 ± 0.02 (range: 0.02–0.07); *T. payamense* sp. nov. (two specimens) = 0.00; *T. resimus* = 0.04 ± 0.03 (range: 0.00–0.07); and *T. uncinatus* = 0.05 ± 0.02 (range: 0.00–0.08). The mean interspecific sequence divergence within the *T. allevatus* group (all subgroups included) was 0.11 ± 0.02 (range: 0.01–0.15). The mean interspecific sequence divergence within the *T. opinatus* subgroup was: 0.09 ± 0.02 (range: 0.01–0.13). The mean interspecific sequence divergence in the *T. opinatus* subgroup without *T. payamense* sp. nov. = 0.09 ± 0.03 (range: 0.01–0.13). The mean interspecific sequence divergence of *T. payamense* sp. nov. *vs* other species in the *T. opinatus* subgroup = 0.08 ± 0.02 (range: 0.05–0.12). The mean interspecific sequence divergence of *T. payamense* sp. nov. *vs* other species in the *T. allevatus* group = 0.11 ± 0.03 (range: 0.05–0.14).

The ML and BI trees (COI and 16S rRNA separately, as well as COI + 16S rRNA combined) were largely congruent with respect to the well-supported nodes (by visual

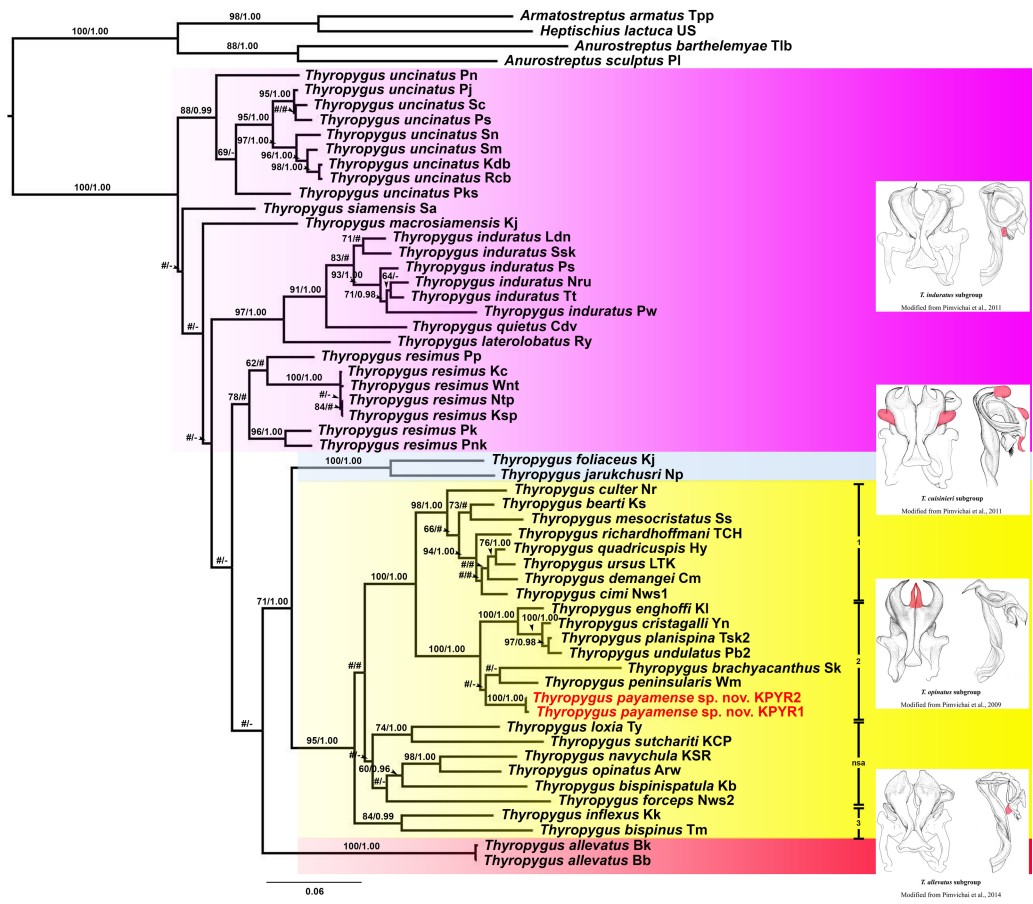

**Figure 1** **Phylogenetic relationships of *Thyropygus* species based on maximum likelihood analysis (ML) and Bayesian Inference (BI) of 1,147 bp in the combined COI + 16S rRNA alignment.** Numbers at nodes indicate node support based on bootstrapping (ML)/posterior probabilities (BI). Scale bar = 0.06 substitutions/site. # indicates nodes with < 50% bootstrap support and < 0.95 posterior probability. - indicates non-supported nodes. The colored areas mark the *T. induratus* subgroup (purple), *T. cuisinieri* subgroup (blue), *T. opinatus* subgroup (yellow), and *T. allevatus* subgroup (red). The vertical lines indicate different clades within the *T. opinatus* subgroup. Abbreviations after species names refer to the locality names in Table S1. Unique gonopodal characters shared by members of each subgroup are highlighted in red. These characters represent key morphological features that define and differentiate the subgroups.

inspection). So, for further discussion, the combined COI + 16S rRNA tree will be used (Fig. 1), while the separate COI and 16S rRNA trees are provided in Figs. S1 and S2.

*Thyropygus payamense* sp. nov. was firmly positioned within the *T. opinatus* subgroup (Fig. 1), whose monophyly was strongly supported (BV = 95; PP = 1.00). The *T. opinatus* subgroup was further divided into a non-supported assemblage (nsa) of six species, *viz.*, *T. bispinispatula*, *T. forceps*, *T. loxia*, *T. navychula*, *T. opinatus*, and *T. sutchariti* (Fig. 1: nsa) and three well-supported clades (Fig. 1: 1–3):

**Clade 1**: was almost maximally supported (BV = 98, PP = 1.00) and comprised eight species from southern Thailand: *T. bearti*, *T. cimi*, *T. culter*, *T. demangei*, *T. mesocristatus*,

*T. quadricuspis*, *T. richardhoffmani*, and *T. ursus*. This clade was maximally supported as sister group of clade 2.

**Clade 2** : was maximally supported (BV = 100, PP = 1.00) and comprised seven species from southern Thailand: *T. brachyacanthus*, *T. cristagalli*, *T. enghoffi*, *T. payamense* sp. nov., *T. peninsularis*, *T. planispina*, and *T. undulatus*.

**Clade 3**: was well-supported (BV = 84, PP = 0.99) and comprised two singleton species from northern, central and western Thailand: *T. inflexus* and *T. bispinus*. The sister group position of this clade was not well-resolved.

Additionally, the *T. cuisinieri* subgroup was well-supported (BV = 99, PP = 1.00), consisting of two singleton species: *T. foliaceus* and *T. jarukchusri*, that jointly were well-supported as sister taxon of the *T. opinatus* subgroup. The *T. allevatus* subgroup was only represented by its nominal species, whose sister group position was not resolved. There was no support for the monophyly of the *T. induratus* subgroup (Fig. 1: assemblage marked in purple).

In the separate COI tree (Fig. S1), clades 1 and 2 were each well-supported, but their sister group relation was not, while clade 3 was only well-supported in the BI analysis, but its sister group relationship was unresolved. In contrast, clade 1 was not supported in the separate 16S rRNA tree (Fig. S2), while clades 2 and 3 were only well-supported in the ML analysis. Nevertheless, the species of clades 1 and 2 were grouped together in a well-supported overarching clade, while the sister group relationship of clade 3 was unresolved. The six species from the non-supported assemblage in the combined tree, remained as such in either of the separate trees since they appeared scattered throughout the *T. opinatus* subgroup. The *T. cuisinieri* subgroup was consistently well-supported by the separate COI and 16S rRNA trees, but its sister group relationships were not. Also the sister group position of *T. allevatus* remained unresolved, while there was no support for the monophyly of the *T. induratus* subgroup.

## Taxonomy

Class Diplopoda de Blainville in Gervais, 1844
Order Spirostreptida Brandt, 1833
Suborder Spirostreptidea Brandt, 1833
Family Harpagophoridae Attems, 1909
Genus *Thyropygus* Pocock, 1894
Informal taxon *Thyropygus allevatus* group sensu *Hoffman (1975)*
Informal taxon *Thyropygus opinatus* subgroup sensu *Pimvichai et al. (2016)*

**Diagnosis.** A subgroup of the *T. allevatus* group. Differing from the *T. induratus*, *T. cuisinieri* and *T. allevatus* subgroups by having an additional projection on the anterior coxal fold (*amp*).

**Included species:**
*T. bearti* Pimvichai, Enghoff & Panha, 2009
*T. bifurcus* (Demange, 1986)
*T. bispinispatula* Pimvichai, Enghoff & Panha, 2009

*T. bispinus* Pimvichai, Enghoff & Panha, 2009

*T. brachyacanthus* Pimvichai, Enghoff & Panha, 2009

*T. casjeekeli* Pimvichai, Enghoff & Panha, 2009

*T. chelatus* Pimvichai, Enghoff & Panha, 2009

*T. cimi* Pimvichai, Enghoff, Panha & Backeljau, 2016

*T. cristagalli* Pimvichai, Enghoff & Panha, 2009

*T. culter* Pimvichai, Enghoff, Panha & Backeljau, 2016

*T. demangei* Pimvichai, Enghoff & Panha, 2009

*T. enghoffi* (Demange, 1989)

*T. erectus* Pimvichai, Enghoff & Panha, 2009

*T. floweri* Demange, 1961

*T. forceps* Pimvichai, Enghoff, Panha & Backeljau, 2016

*T. implicatus* Demange, 1961

*T. inflexus* (Demange, 1989)

*T. loxia* Pimvichai, Enghoff & Panha, 2009

*T. mesocristatus* Pimvichai, Enghoff, Panha & Backeljau, 2016

*T. navychula* Pimvichai, Enghoff, Panha & Backeljau, 2016

*T. opinatus* (Karsch, 1881)

*T. payamense* sp. nov.

*T. peninsularis* Hoffman, 1982 (see Discussion)

*T. planispina* Pimvichai, Enghoff, Panha & Backeljau, 2016

*T. quadricuspis* Pimvichai, Enghoff & Panha, 2009

*T. richardhoffmani* Pimvichai, Enghoff & Panha, 2009

*T. sutchariti* Pimvichai, Enghoff, Panha & Backeljau, 2016

*T. undulatus* Pimvichai, Enghoff, Panha & Backeljau, 2016

*T. ursus* Pimvichai, Enghoff, Panha & Backeljau, 2016

## Species description

### *Thyropygus payamense* sp. nov. (Figs. 2–4)

**Material examined.** Holotype male (CUMZ-D00155), THAILAND, Ranong Province, Muang Ranong District, Payam Island, Aow Yai, 10 m a.s.l., 9°43′45″N, 98°23′25″E, 13/11/2022, leg. P. Pimvichai, T. Backeljau, B. Segers, K. Breugelmans and S. Saratan. Paratypes five males (CUMZ-D00155-1), eight females (CUMZ-D00155-2), same data as holotype, one male (NHMD 1184744) Thailand, Ranong Province, Muang Ranong District, Payam Island, /04/2013, leg. J. Urbanski.

**Etymology.** The name refers to Payam Island, the type locality of this species.

**Diagnosis.** A species of the *T. opinatus* subgroup in the *T. allevatus* group. Differs from all other species of the *T. opinatus* subgroup by having (1) a small, slender, pointed spine (*sfe*) at base of femoral spine (*fe*), (2) the mesal process of anterior coxal fold (*amp*) short, forming a triangular process, and (3) tibial spine (*ti*) short, slender, slightly curving mesad.

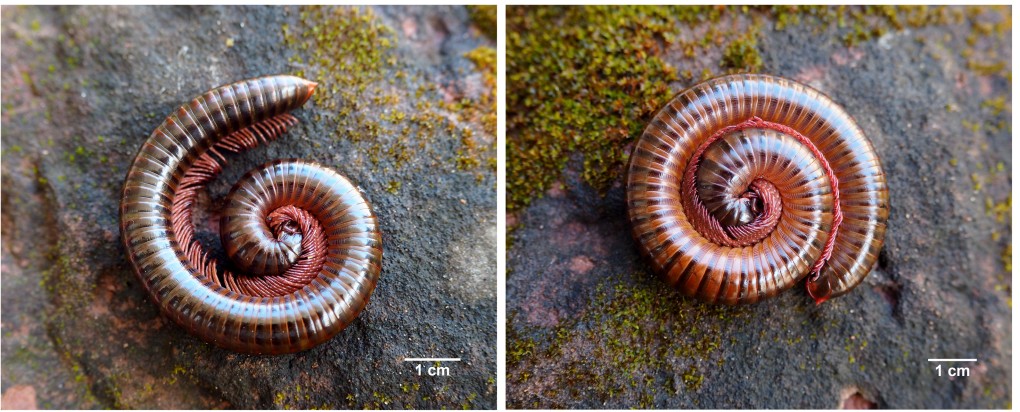

**Figure 2** Live *Thyropygus payamense* sp. nov., paratype, male (CUMZ - D00155-1).

**Description.** Adult males with 60–61 podous rings, no apodous rings. Length 13–14 cm, width 8.6–9.3 mm. Adult females with 60–62 podous rings, no apodous rings. Length 12–14 cm, width 8.7–9.4 mm.

**Colour.** Overall colour of living animal (Fig. 2) dark brown. Antennae, legs, epiproct, paraprocts and hypoproct reddish brown.

**Gonopods** (Figs. 3A–3D). Anterior coxal fold (*ac*; Fig. 3A): the lateral process (*alp*) flattened and broad, apically curved caudad and terminating in a short spine, the lateral margin slightly folded; the mesal process (*amp*) broad at base, apically gradually narrowed, pointed, forming a triangular process, $\frac{1}{4}$ of the height of the lateral process (*alp*). Posterior coxal fold (*pc*; Fig. 3B) basally with moderately high paracoxites (*px*), forming shelf to accommodate telopodite, distally with two processes: mesal process (*pmp*) very small, directed distolaterad; lateral process (*plp*) digitiform, directed distad. Telopodite (Figs. 3C–3D) leaving coxite over shelf of posterior coxal fold; the femoral spine (*fe*) very long, slender, curving backward, with a small, slender, pointed spine (*sfe*) at its base, *in situ* resting behind *alp*; the tibial spine (*ti*) short, slender, slightly curving mesad; the apical part: spatulate lobe (*sl*) small, rounded; palette (*pa*) simple, gutter-like; distally with about 11 brownish blepharochaetae (*bp*).

**DNA barcodes.** The GenBank accession number of the COI barcode of the holotype is PV019345 and that of 16S rRNA is PV029246 (voucher code CUMZ-D00155). The COI barcodes of paratypes are PV019346–PV019347 (voucher code CUMZ-D00155-1 for a male and voucher codes CUMZ-D00155-2, CUMZ-D00155-2-1 for 2 females). The 16S rRNA barcode of the paratype with voucher code CUMZ-D00155-2 is PV029247.

**Distribution.** The species is known only from its type locality in Ranong Province, Thailand (Fig. 4). It was collected in Aow Yai, where the specimens were found crawling and hiding underneath leaf litter of coconut trees, jackfruit trees, and other native vegetation.

**Key to the 29 currently recognized species of the *T. opinatus* subgroup; figures underneath a couplet illustrate the relevant gonopodal characteristics referred to in the couplet (updated from *Pimvichai et al., 2016*) (Supplementary Files)**

1. Apical part of telopodite with spatulate lobe (*sl*)……………………............... …………....2
— Apical part of telopodite with lamellar lobe (*ll*)…………….………...............…………22
2. Spatulate lobe (*sl*) distally drawn out into one or two sharp dark brown spine(s)……..………….…….……………….……….……………….…....................3
— Spatulate lobe (*sl*) distally expanded and/or rounded, spoon-like, without a spine ....…..............……………….…………………….……………….…………………….………9

## DISCUSSION

Morphologically, *Thyropygus payamense* sp. nov. undoubtedly belongs to the genus *Thyropygus*, as it has the diagnostic characteristics of the genus listed by *Pimvichai, Enghoff & Panha (2009a)*. These include: (1) body rings that are not strongly wrinkled dorsally, (2) ozopores begining on body ring 6, (3) very long stigmatic grooves, (4) ventral soft pads on the postfemur and tibia of male walking legs, (5) a triangular gonopod sternum, (6) a gonopod telopodite with a femoral spine and often a tibial spine, (7) a prostatic groove terminating apically on a solenomere or prostatic lobe (apical palette of the telopodite), and (8) a voluminous apical palette that is more or less expanded and forms a gutter-like structure. Within the genus *Thyropygus*, *T. payamense* sp. nov. belongs to the *T. allevatus* group because it has a tibial and a femoral spine on the gonopod telopodite, with the tibial spine being notably long and recurved proximally toward the femoral spine. Finally, it is assigned to the *T. opinatus* subgroup because it has an additional projection on the anterior coxal fold.

The mean interspecific DNA sequence divergence values of *T. payamense* sp. nov. relative to other species in the *T. allevatus* group (mean values: 0.13 for COI and 0.11 for 16S rRNA) or the *T. opinatus* subgroup (mean values: 0.11 for COI and 0.08 for 16S rRNA) support the species-level distinction of *T. payamense* sp. nov. since they are of a comparable magnitude as the mean interspecific divergences for other species pairs in this group and subgroup (mean values: 0.12 for COI and 0.09 for 16S rRNA in the *T. induratus* subgroup; mean values: 0.11 for COI and 0.09 for 16S rRNA in the *T. cuisinieri* subgroup). The mean interspecific COI divergence values of *T. payamense* sp. nov. also align well with those observed in some genera of spirobolidan families, such as Pseudospirobolellidae with *Coxobolellus* Pimvichai, Enghoff, Panha & Backeljau, 2020 (mean 0.11; range: 0.06–0.15) (*Pimvichai et al., 2020*) and *Siliquobolellus* Pimvichai, Enghoff, Panha & Backeljau, 2022 (mean: 0.12; range: 0.08–0.15) (*Pimvichai et al., 2022*) or Pachybolidae with *Atopochetus* Attems, 1953 (mean: 0.14; range 0.09–0.17) and *Litostrophus* Chamberlin, 1921 (mean: 0.11; range 0.09–0.11) (*Pimvichai et al., 2018*).

The combination of its comparative DNA sequence divergence values, its phylogenetic placement as a well-supported clade, and its gonopodal differentiation, provide a solid

3. Spatulate lobe (*sl*) terminating in two sharp brown spines, the outer spine slightly smaller and shorter than the inner one; lateral process of anterior coxal fold (*alp*) slender, slightly curving mesad; mesal process of anterior coxal fold (*amp*) almost as long as *alp*, flattened............................................................................**T. bispinispatula**

— Spatulate lobe (*sl*) terminating in a single sharp dark brown spine.....................................4

4. Telopodite without a lobe distal to *fe*; lateral process of anterior coxal fold (*alp*) long, slender, regularly curved, tip close to tip of opposite *alp*, the two together forming a circle; mesal process of anterior coxal fold (*amp*) straight, shorter than *alp*; femoral spine (*fe*) directed distad,

— Telopodite distally to *fe* with a large, round lobe (*lo*) projecting distolaterally.................5

5. Lateral process of anterior coxal fold (*alp*) very slender, regularly curved..........................6

— Lateral process of anterior coxal fold (*alp*) different, broader and/or with several apical denticles.............................................................................................................................8

6. Mesal margin of lateral process of anterior coxal fold (*alp*) with fine serrations; mesal process of anterior coxal fold (*amp*) almost as long as *alp*, broadly expanded, apically sharp, straight distad, mesal margin forming a strong longitudinal crest (*lc*) in posterior

— Mesal margin of lateral process of anterior coxal fold (*alp*) without serrations, tip of lateral process close to tip of the opposite side, the two together forming a circle...................................................................................................... .7

7. Mesal process of posterior coxal fold (*pmp*) strongly developed along anterior-posterior axis........................................................................................................**T. floweri**

— Mesal process of posterior coxal fold (*pmp*): slender, directed distolaterad.......**T. forceps**

8. Lateral process of anterior coxal fold (*alp*) broad, apically gradually narrowed; mesal process of anterior coxal fold (*amp*) almost as long as lateral process (*alp*), slender, straight, terminally

— Lateral process of anterior coxal fold (*alp*) apically bent abruptly mesad, tip with serrate margins; mesal process of anterior coxal fold (*amp*) much shorter than lateral process (*alp*), directed mesodistad, simple, pointed; mesal process of posterior coxal fold (*pmp*): strongly developed along anterior-posterior axis............................................**T. implicatus**

9. Telopodite with a single femoral spine (*fe*)...................................................... ...10

— Telopodite with two femoral spines (*fe 1* and *fe 2*)...................................................19

10. Mesal process of anterior coxal fold (*amp*) short...................................................11

— Mesal process of anterior coxal fold (*amp*) long, slender ...........................................13

11. Telopodite with slender tibial spine (*ti*), not curving mesad; *fe* curving backward, without small spine; mesal process of anterior coxal fold (*amp*) very short, pointed......**T. peninsularis**

— Telopodite with short, slender tibial spine (*ti*), curving mesad ....................12

12. Femoral spine (*fe*) with a small, slender, pointed spine (*sfe*) at base (Fig. 3C); mesal process of anterior coxal fold (*amp*) short, forming a triangular process; telopodite distally to *fe* without a small round lobe (*lo*)............................................**T. payamense**  **sp. nov.**

— Femoral spine (*fe*) without a small slender, pointed spine (*sfe*) at base; telopodite distally to *fe* with a small round lobe (*lo*) projecting distolaterally...............................................**T. loxia**

13. Lateral process of anterior coxal fold (*alp*) apically abruptly truncate...........  **T. bearti**

— Lateral process of anterior coxal fold (*alp*) apically pointed...........................................14

14. Mesal process of anterior coxal fold (*amp*) shorter than lateral process (*alp*)...............15

— Mesal process of anterior coxal fold (*amp*) as long as lateral process (*alp*)...............16

15. Mesal process of anterior coxal fold (*amp*) directed obliquely distomesad, slender, straight..................................................................................................... **T. chelatus**

— Mesal process of anterior coxal fold (*amp*) directed distad, thicker, slightly sigmoid................................................................................................  **T. brachyacanthus**

16. Mesal process of anterior coxal fold (*amp*) directed obliquely distomesad, tip overlapping tip of opposite *amp*; lateral process of posterior coxal fold (*plp*) a massive, broad lobe, projecting laterad…..………………………………..........................……**T. sutchariti**

— Mesal process of anterior coxal fold (*amp*) directed distad .... ….....…...……………….…...17

17. Lateral process of anterior coxal fold (*alp*) apically without a crest; telopodite distally with a rounded lobe (*lo*); margins of spatulate lobe (*sl*) terminally meeting in a distinct angle .........................................................................................................................**T. bispinus**

— Lateral process of anterior coxal fold (*alp*) apically with a crest.……………..……..……….…18

18. Mesal process of anterior coxal fold (*amp*) apically irregularly tuberculate; telopodite distally without a rounded lobe (*lo*)................................................. ...…...................**T. inflexus**

— Mesal process of anterior coxal fold (*amp*) slender, straight, its tip pointed, its mesal margin forming a strong longitudinal crest (*lc*) in posterior view……**T. mesocristatus**

19. Anterior coxal fold (*ac*) with an additional spine-like process (*aip*) between *alp* and *amp*; lateral process of anterior coxal fold (*alp*) broad, mesal margin concave, tip with serrate margins, chicken comb-like; mesal process of anterior coxal fold (*amp*) much shorter than lateral process (*alp*), directed mesodistad, simple, pointed; both femoral spines (*fe*) slender, long..................................................................................................................**T. cristagalli**

— Anterior coxal fold (*ac*) without an additional spine-like process (*aip*) between *alp* and *amp*.…………………………………………………………...……………….…………......20

20. Lateral process of anterior coxal fold (*alp*) apically without a crest, flattened, slightly curved, its laterodistal margin coarsely dentate, terminating in a short, sharp, pointed spine; mesal process (*amp*) much shorter than *alp*, directed distad, tip curving mesad, pointed; both femoral spines (*fe 1, fe 2*) long, curving backward; tibial spine (*ti*) long, not curving in horizontal plane.…………………………………………………...…….....……………… **T. culter**

— Lateral process of anterior coxal fold (*alp*) apically with a crest extending caudad.…………………………………………………………………………………..…21

21. Lateral process (*alp*) flattened, curving mesad, laterodistal margin coarsely dentate, terminating in a short spine, tip curving against the tip of opposite side; mesal process (*amp*) much shorter than *alp*, slender, curving mesad; both femoral spines (*fe 1, fe 2*) broad, long; tibial spine (*ti*) long, curving in horizontal plane, not ending in a sharp spine...………**T. undulatus**

— Lateral process (*alp*) regularly curved, terminating in a sharp, slightly upward pointing spine; mesal process (*amp*) slightly shorter than *alp*, flattend, straight, directed distad; tibial spine (*ti*) flattend, short, curving mesad…………………………**T. planispina**

22. Telopodite with a single femoral spine.…………...………………………...…...…………….23

— Telopodite with two femoral spines.……………………………………….........……………25

23. Lateral process of anterior coxal fold (*alp*) without an apical crest; mesal process of anterior coxal fold (*amp*) shorter than and as broad as *alp*, directed distad; femoral spine (*fe*) very long and slender...............................................................................**T. casjeekeli**

— Lateral process of anterior coxal fold (*alp*), with a sharp crest on the posterior surface near the tip.………………………………………………………..……………...……….24

24. Lateral process of anterior coxal fold (*alp*) flattened, slightly curved, inflexed; femoral spine (*fe*) very long, slender, with an additional lamella at base………… **T. quadricuspis**

— Lateral process of anterior coxal fold (*alp*) regularly curved, basally broad, gradually tapering towards end and ending in sharp point; femoral spine (*fe*) very long, slender, without an additional lamella at base…..........…………………...……....................................**T. cimi**

25. Lateral process of anterior coxal fold (*alp*) flatten, broad..………………..………………....26
— Lateral process of anterior coxal fold (*alp*) slender, regularly curved, sickle-shaped ……………………...………...……’…...……..……………….…………………………………....27
26. Lateral process of anterior coxal fold (*alp*) terminating in a very short external spine and a very long internal one; mesal process of anterior coxal fold (*amp*) as long as *alp*; first femoral spine (*fe 1*) very short, pointed; second femoral spine (*fe 2*) very long, as long as tibial spine (*ti*); an additional lamella at both side of base of *fe 2*.........………………...***T. richardhoffmani***
— Lateral process (*alp*) flattened, apically curved laterad as a short spine, lateral margin of *alp* slightly folded; mesal process (*amp*) shorter than *alp*, slender, straight, directed distad, pointed; the first femoral spine (*fe 1*) very short, directed upward, situated above *fe 2*, the second *fe* (*fe2*) very long, slender, curved downward.………………………………………………***T. ursus***
27. Mesal margin of lateral process of anterior coxal fold (*alp*) simple, without a caudad spine or crest; mesal process of anterior coxal fold (*amp*) much shorter than lateral process (*alp*), curved, pointed.……………………………………………...……………...... ***T. enghoffi***
— Mesal margin of lateral process of anterior coxal fold (*alp*) with a caudad small spine or crest...........................................................................................................................................28
28. Mesal margin of lateral process of anterior coxal fold (*alp*) with a small caudad crest; mesal process of anterior coxal fold (*amp*) slightly shorter than *alp*, slightly sigmoid, pointed.………………………...……………...............……………………. ***T. bifurcus***
—Mesal margin of lateral process of anterior coxal fold (*alp*) with a short curved caudad spine; mesal process of anterior coxal fold (*amp*) as long as *alp*, straight…....................***T. demangei***

basis to recognize *T. payamense* sp. nov. as a well-defined, separate species that complies at least with the morphological, biological, phylogenetic and lineage species concepts.

The addition of *Thyropygus payamense* sp. nov. (and *T. peninsularis*; see further below) to the *T. opinatus* subgroup did not affect the strong support for the monophyly of this subgroup, which now comprises 29 species. Hence, the congruence between morphological and DNA sequence data in the *T. opinatus* subgroup seems to be consistent and robust. It suggests that the defining, shared characters of this multi-species subgroup represent true synapomorphies. This contrasts sharply with the phylogenetic interpretation of the *T. induratus* subgroup, which was recently questioned because the discovery of two new species that morphologically clearly belong to this subgroup (*T. panhai* and *T. somsaki*) obliterated the support of its monophyly as inferred by COI sequence data. Hence the congruence between the morphological and DNA sequence data for the *T. induratus* subgroup was disrupted (*Pimvichai, Enghoff & Backeljau, 2023*).

The three clades within the *T. opinatus* subgroup identified in this study jointly form Clade 1A3 described by *Pimvichai et al. (2016)*, with the inclusion of *T. payamense* sp. nov. and *T. peninsularis*. It is striking that the Thai members of the *T. opinatus* subgroup only occur in southern Thailand (Clades 1, 2, and nsa), except for the two species of clade 3, which are distributed in northern, central and western Thailand. Conversely, no species from the other subgroups of the *T. allevatus* group were hitherto found in southern Thailand.

Southern Thailand, part of the Sundaland biogeographic region, is characterized by a unique mix of fauna influenced by its peninsular geography, tropical climate, and historical land connections to surrounding regions (*Parnell, 2013*). As such, the present data tentatively suggest that *T. opinatus* subgroup clades 1, 2 and the nsa jointly may

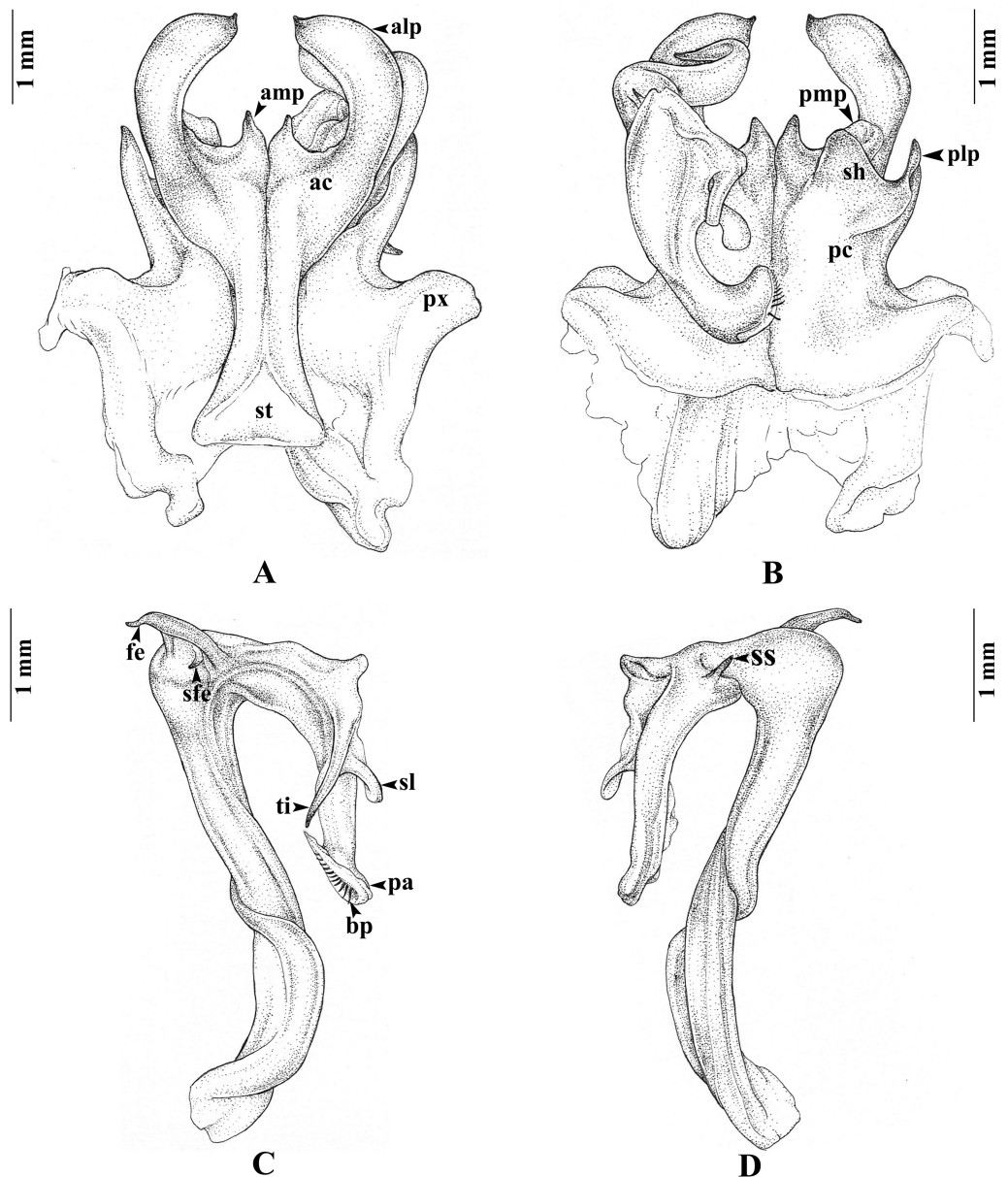

**Figure 3** *Thyropygus payamense* **sp. nov., holotype, gonopods (CUMZ - D00155).** (A) Anterior view, left telopodite removed. (B) Posterior view, left telopodite removed. (C) Left telopodite, posterior-mesal view, note the small, slender, pointed spine (*sfe*). (D) Left telopodite, anterior-lateral view.

represent an endemic species radiation in the peninsular area of Thailand, Malaysia and Myanmar. Yet, further phylogeographic analyses incorporating a broader sampling of populations, taxa and DNA markers are needed to infer the precise evolutionary and biogeographical history of these species.

*Thyropygus peninsularis* was initially suggested to belong to the *T. erythropleurus* group by *Hoffman (1982)*, because it has no recurved tibial spine proximally directed towards the femoral spine—a defining feature of the *T. allevatus* group. Therefore, *Pimvichai,*

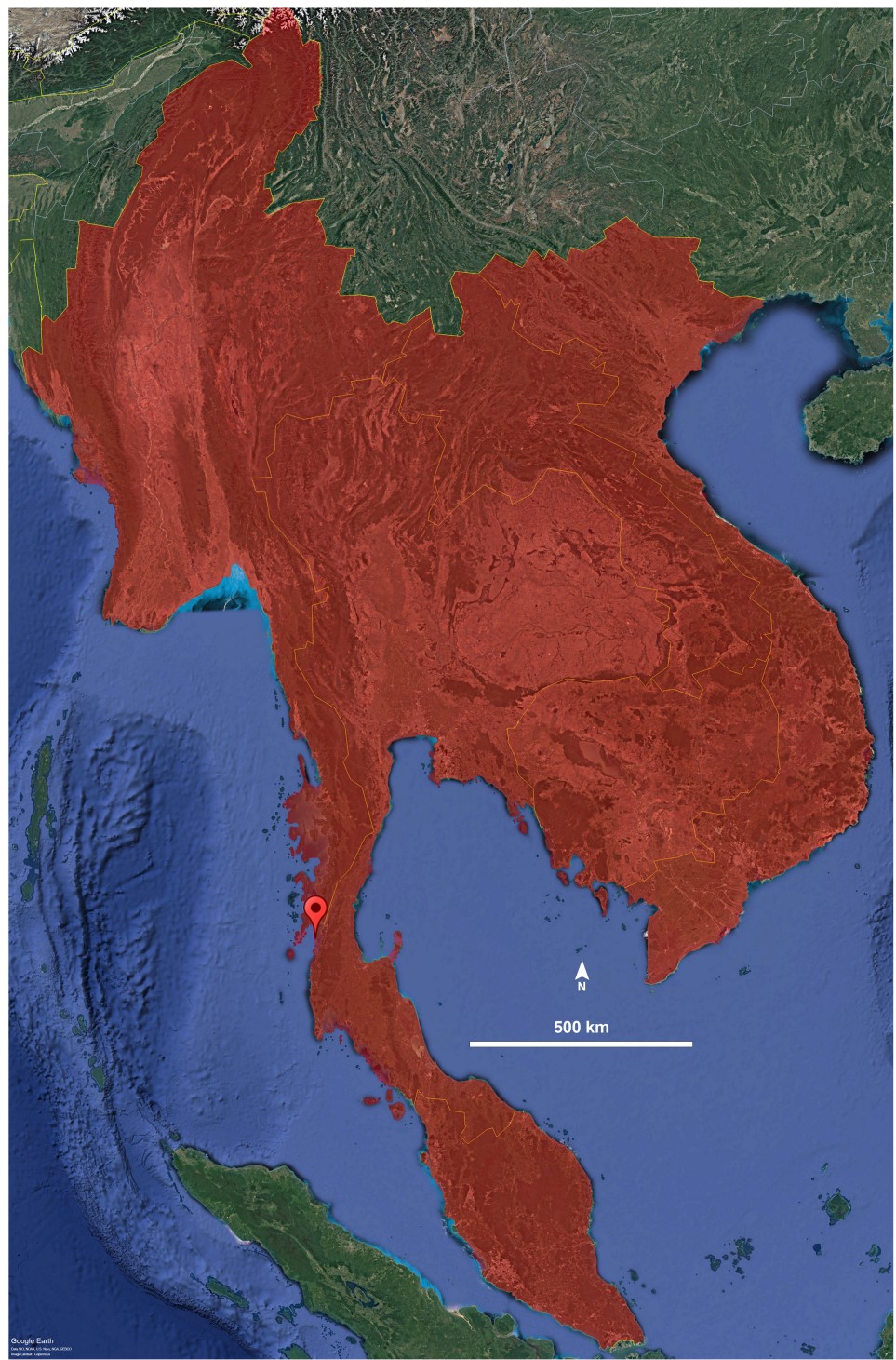

**Figure 4** **Distribution of the genus *Thyropygus*.** Droplet indicates the type locality of *T. payamense* sp. nov. Map generated using Google Earth Pro (Version 7.3.6.9796).

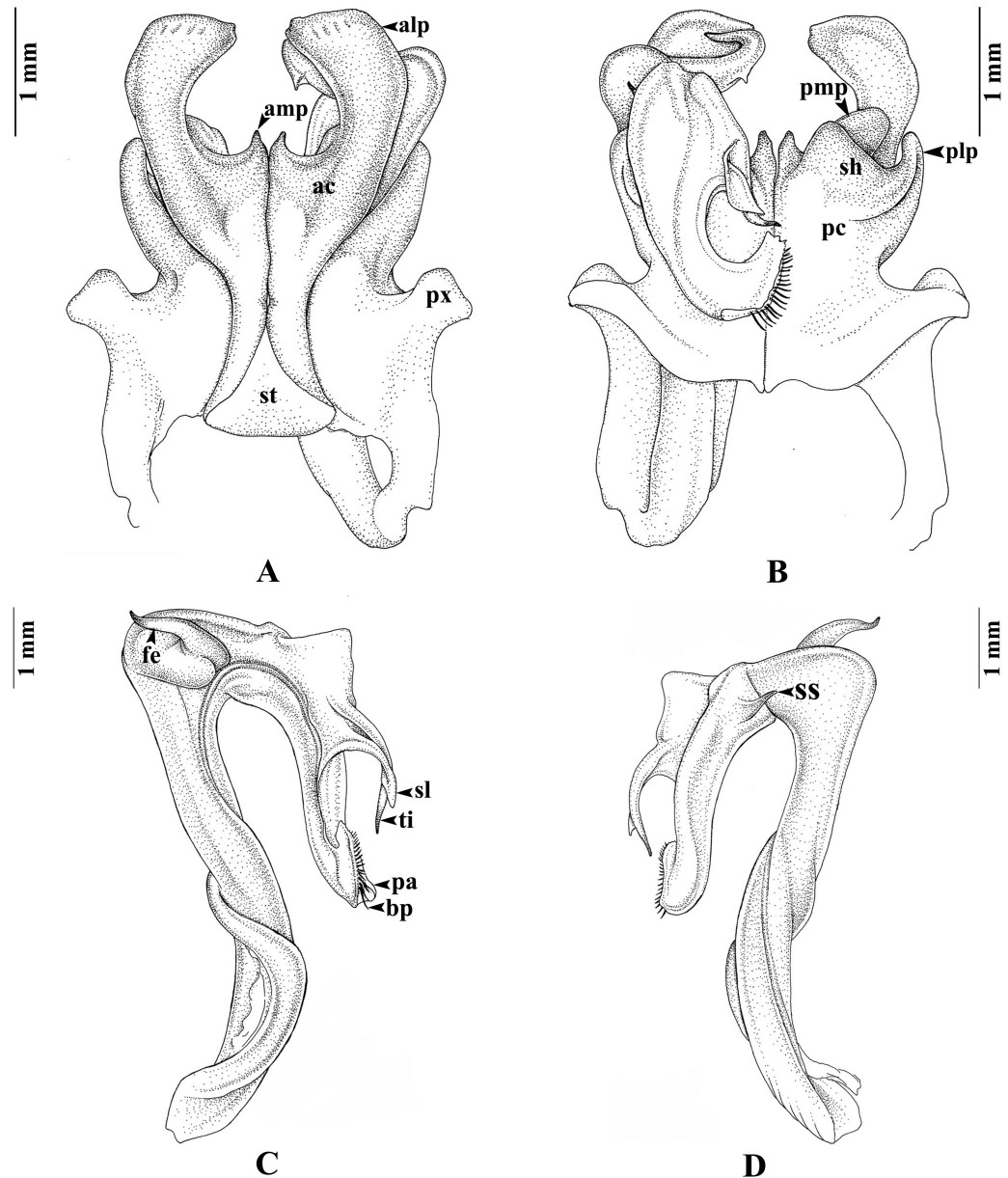

**Figure 5** *Thyropygus peninsularis,* **gonopods.** *Thyropygus peninsularis*, specimen from Wang-Matcha, Kapoe, Ranong, Thailand, gonopods (CUMZ - D00011). (A) Anterior view, left telopodite removed. (B) Posterior view, left telopodite removed. (C) Left telopodite, posterior-mesal view. (D) Left telopodite, anterior-lateral view (modified from *Panha, Pimvichai & Enghoff, 2009*).

*Enghoff & Panha (2009a)* followed *Hoffman (1982)* and did not include *T. peninsularis,* in the *T. allevatus* group. However, *T. peninsularis* possesses a small spatulate lobe at the apical part of the telopodite, along with a very short additional mesal projection on the gonopod's anterior coxal lobe (Fig. 5), similar to *T. loxia.* These features are shared by most species in the *T. opinatus* subgroup. Furthermore, DNA sequence analysis (COI and 16S rRNA) firmly placed *T. peninsularis* within the *T. opinatus* subgroup (*Pimvichai,*

*Enghoff & Panha, 2014*, present results). Based on these morphological and DNA sequence data, we formally confirm the assignment of *T. peninsularis* to the *T. opinatus* subgroup, as was implicitly done by *Pimvichai, Enghoff & Backeljau (2023)*. These findings highlight, once more, the importance of integrating morphological and molecular data for resolving and/or re-interpreting taxonomic ambiguities.

## CONCLUSIONS

While the support for the monophyly of some species subgroups within the *Thyropygus allevatus* group disappears by increased species sampling, the high support for the monophyly of the *T. opinatus* subgroup remains unaffected after increased species sampling by the inclusion of (1) *T. payamense* sp. nov., described in this study, and (2) *T. peninsularis*, a species formerly assigned to the *T. erythropleurus* group, but for which DNA sequence data and a re-interpretation of its gonopod morphology show that it actually belongs to the *T. opinatus* subgroup. As a consequence, the congruence between the DNA sequence data and the defining synapomorphies in gonopod morphology remains consistent and robust in the *T. opinatus* subgroup, which now comprises 29 species. While it is too early to draw firm phylogeographic conclusions, these data tentatively suggest that with the exception of *T. bispinus* and *T. inflexus*, the *T. opinatus* subgroup may represent an endemic species radiation in the peninsular area of Thailand, Malaysia and Myanmar. Finally, the results illustrate the importance of combining further species sampling with integrative research to resolve taxonomic ambiguities and explore evolutionary relationships in these millipedes.

## ACKNOWLEDGEMENTS

Sathit Saratan (Sirindhorn Museum, Thailand) is warmly acknowledged for his great assistance during fieldwork. We are indebted to Thita Krutchuen (College of Fine Arts, Bunditpatanasilpa Institute, Thailand) for the excellent drawings. We thank Zoltán Korsós (Hungarian Natural History Museum) and another, anonymous, reviewer for their helpful comments on the manuscript.

### Funding
This research project was financially supported by Mahasarakham University. The funders had no role in study design, data collection and analysis, decision to publish, or preparation of the manuscript.

### Grant Disclosures
The following grant information was disclosed by the authors:
Mahasarakham University.

### Competing Interests
The authors declare there are no competing interests.

## Author Contributions

- Piyatida Pimvichai conceived and designed the experiments, performed the experiments, analyzed the data, prepared figures and/or tables, authored or reviewed drafts of the article, and approved the final draft.
- Henrik Enghoff performed the experiments, analyzed the data, authored or reviewed drafts of the article, and approved the final draft.
- Karin Breugelmans analyzed the data, prepared figures and/or tables, and approved the final draft.
- Brigitte Segers analyzed the data, prepared figures and/or tables, and approved the final draft.
- Thierry Backeljau performed the experiments, analyzed the data, authored or reviewed drafts of the article, and approved the final draft.

## Field Study Permissions

The following information was supplied relating to field study approvals (i.e., approving body and any reference numbers):

This research was conducted under the approval of the Animal Care and Use regulations (numbers U1-07304-2560) by the National Research Council of Thailand.

## DNA Deposition

The following information was supplied regarding the deposition of DNA sequences:

The new COI and 16S rRNA sequences are available at GenBank: PV019345–PV019347 and PV029246–PV029247.

## Data Availability

The estimates of COI, 16S rRNA, and COI+16S rRNA sequence divergences within and among *Thyropygus* species and outgroup taxa are available in the Supplementary Files.

## New Species Registration

The following information was supplied regarding the registration of a newly described species:

Publication LSID: urn:lsid:zoobank.org:pub:68E7FD7F-A8E3-4BE9-9B4B-136CDEEBEA88 *Thyropygus payamense* sp. nov. LSID: urn:lsid:zoobank.org:act:20E6E0CA-A4E5-4A51-9FB6-2CF07C301883.

## Supplemental Information

Supplemental information for this article can be found online at http://dx.doi.org/10.7717/peerj.19277#supplemental-information.

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
