# Peer review of "Morphological and DNA sequence data uncover a new millipede species in the Thyropygus opinatus subgroup and assign T. peninsularis to this subgroup (Diplopoda: Spirostreptida: Harpagophoridae)"

_PeerJ, doi:10.7717/peerj.19277_

## Round 0.1 · original submission · Minor Revisions

Dear authors, I ask you to carefully correct the manuscript in accordance with the reviewers' fundamental comments and hope that the new version of this article can be published.

·

Basic reporting

It is a clear, professional manuscript with all the necessary details to support the new species description. Well-written in ENglish (with some small typographical errors marked in the uploaded version).
I checked one reference to a Hungarian journal, which is probably mistakenly cited: all other references should be carefully checked by the authors again.
Figures are beautiful, phylogenetic diagrams are easily representative.

Experimental design

Everything looks fine.

Validity of the findings

The results are supported, and soundly valid. All necesary data have been provided. Conclusions are understandable and correct.

Additional comments

No additional comments.

Reviewer 2 ·

Basic reporting

In this article, the author describes a new species and provides partial sequences of COI and 16S. From a morphological perspective, I find no issues. It is commendable that the author has included corresponding figures in the key, which enhances clarity and intuitiveness. However, a few minor issues require attention or clarification.

Experimental design

no comment

Validity of the findings

no comment

Additional comments

a few minor issues require attention or clarification.
1、the article layout requires adjustment, as lines 41, 56, and 66 exhibit inconsistencies.
2、the author constructed a phylogenetic tree utilizing COI and partial 16S sequences; however, the sampling was insufficient, resulting in a considerable number of nodes with support rates below 50% or lacking support altogether. Is it feasible to attempt constructing a tree based on morphological characteristics?
3、a substantial portion of the article is dedicated to discussing genetic distances. Could the authors provide a summary reference value for the genetic distances both within and between Thyropygus species?
4、Lines“(2) T. peninsularis, a species formerly assigned to the T. erythropleurus group, but for which DNA sequence data and a re-interpretation of its gonopod morphology show that it actually belongs to the T. opinatus subgroup. ”,but I can not find T. erythropleurus in Figure 1。
5、To enhance intuitiveness, Figure 1 requires further refinement.
6、In Figure 5C, the caption states that T. peninsularis possesses the structure of sfe; however, I was unable to identify this structure, and its location is not indicated in the figure.

---

## Round 0.2 · accepted · Accept

Dear authors, I am pleased to inform you that your article has been accepted for publication.

·

Basic reporting

After reviewing the corrected and revised manuscript, I do not find the need of any further improvement. In my opinion, the manuscript is ready for copy editing and publication.

Experimental design

No comment.

Validity of the findings

All valid.

Additional comments

No comments.

Reviewer 2 ·

Basic reporting

no comment

Experimental design

no comment

Validity of the findings

no comment

Additional comments

no comment